

# Active subglacial lakes in the Canadian Arctic identified by multi-annual ice elevation changes

Whyjay Zheng[1,2], Wesley Van Wychen[3], Tian Li[4], and Tsutomu Yamanokuchi[5]

[1]Center for Space and Remote Sensing Research, National Central University, Taoyuan City 320317, Taiwan
[2]Taiwan Polar Institute, National Central University, Taoyuan City 320317, Taiwan
[3]Department of Geography and Environmental Management, University of Waterloo, Waterloo N2L 3G1, Ontario, Canada
[4]Bristol Glaciology Centre, University of Bristol, Bristol BS8 1SS, United Kingdom
[5]Remote Sensing Technology Center of Japan, Tokyo 105-0001, Japan

**Correspondence:** Whyjay Zheng (whyjz@csrsr.ncu.edu.tw)

**Abstract.** Subglacial lakes influence glacier hydrology, dynamics, and mass balance; however, they are poorly documented outside the polar ice sheets. Here we use high-resolution digital elevation models from 2011 to 2021 and perform regression analysis to characterize 37 subglacial lakes (35 of which are newly identified) across the Canadian Arctic, a region that loses more than 50 gigatonnes of glacier ice yearly. These lakes have an area of 0.3–48.5 km$^2$ and influence surface elevation by 10–150 m, corresponding to a water volume of 0.003–4.5 km$^3$. We classify them into (1) classic subglacial lakes, (2) terminal subglacial lakes at places where two glacier termini converge and coalesce, and (3) partial subglacial lakes with an area of open water at the ice margin. Types 2 and 3 are new and are nearly exclusive in the Canadian Arctic. Lake activities negatively correlate with regional mass balance, implying a need for fine-scale monitoring in the era of accelerated glacier loss.

## 1 Introduction

Subglacial lakes can have significant impacts on subglacial hydrology, ice dynamics, and glacier mass balance. Migration of subglacial lake water can change basal friction and temporarily accelerate ice flow (Magnússon et al., 2007; Stearns et al., 2008; Bell, 2008; Siegfried et al., 2016; Andersen et al., 2023). A subglacial lake can also connect to multiple subglacial channels and other lakes, forming an active hydrological system a few kilometers long (Wingham et al., 2006; Fricker and Scambos, 2009; Flament et al., 2014; Fricker et al., 2014; Siegfried and Fricker, 2018; Hodgson et al., 2022; Andersen et al., 2023). The volume of water transported within the subglacial lake system can be up to 0.1–10 km$^3$ within a year (Wingham et al., 2006; Stearns et al., 2008; Fricker and Scambos, 2009; Flament et al., 2014; Livingstone et al., 2022; Andersen et al., 2023). During subglacial lake drainage, glacier speed can increase by 5–10%, bringing more ice to the downstream area where surface and frontal ablations are strong (Stearns et al., 2008; Siegfried et al., 2016; Andersen et al., 2023). This acceleration can translate to a few additional gigatonnes of ice discharge, suggesting subglacial floods as a controlling factor of glacier mass balance (Stearns et al., 2008).

Despite the importance of subglacial lakes, our understanding of these hidden water bodies is still very limited. This is mainly because our observational history for most glacierized areas is relatively short compared to the decadal evolution of



subglacial systems (Fricker et al., 2014). Although the earliest records of subglacial drainage events date back to the twentieth century in Iceland and Antarctica (Björnsson et al., 2001; Wingham et al., 2006), regional-scale identification of subglacial

lakes was not available until the advance of space-borne Interferometric Synthetic Aperture Radars (InSAR; e.g., ERS-2 and Radarsat) and satellite altimeters (e.g., ICESat and CryoSat-2) within the last twenty years (Livingstone et al., 2022). To date, regional subglacial lake inventories are available for the Greenland Ice Sheet (Bowling et al., 2019), Antarctica (Smith et al., 2009; Wright and Siegert, 2012; Siegfried and Fricker, 2021), and Iceland (Björnsson, 2003). The first global inventory was released in 2022 and contains 773 subglacial lakes (Livingstone et al., 2022). However, most of them are located in Antarctica

(87%) and Greenland (8%). Hydrological model predictions suggest that thousands of subglacial lakes remain to be detected outside the polar ice sheets, including the Canadian Arctic (Livingstone et al., 2013; MacKie et al., 2020; Livingstone et al., 2022).

The Canadian Arctic (Figure 1) is a vast and partially glacierized region that comprises ∼14% of the Earth's glaciers and ice caps by area (Sharp et al., 2011). The northern Canadian Arctic is one of the major contributors to the global glacier mass

loss at a rate of $-30.5 \pm 2.6$ Gt yr⁻¹, which is only preceded by Alaska and the Greenland periphery; the southern Canadian Arctic is ranked fifth ($-23.1 \pm 2.1$ Gt yr⁻¹) among all 19 regions globally (Zemp et al., 2025). Despite its recent mass loss, so far only a few subglacial lakes have been identified in this area. The first discovery was a hypersaline lake beneath the Devon Ice Cap indicated by airborne radar sounding data(Rutishauser et al., 2018), although the latest field data showed no evidence of subglacial water (Killingbeck et al., 2024). The second identification was a large (52 km²) and active subglacial lake and a

smaller one nearby under Manson Icefield, which was just reported in 2024 from analysis of ICESat-2 data and digital elevation models (DEMs) (Gray et al., 2024b). Considering the proximity of these two lakes, it is very likely that other active subglacial lakes exist in the vast Canadian Arctic (Gray et al., 2024b).

When a subglacial lake recharges and drains, the overlying ice layer gains and loses support from subglacial water. This process results in vertical deflection of the ice surface, which can be detected by satellite geodesy (Björnsson et al., 2001;

Björnsson, 2003; Willis et al., 2015; Gray et al., 2024a). Spaceborne altimeters such as ICESat-2 and CryoSat-2 have helped identify numerous subglacial lakes (Gray et al., 2024b; Fan et al., 2023; Stubblefield et al., 2023; Siegfried et al., 2023). However, this method requires a satellite swath to pass the lake area, limiting the potential discovery for kilometer-scaled lakes outside satellite swaths (Khalsa et al., 2022; Sørensen et al., 2024). Recent public releases of high-resolution and multitemporal DEMs (Porter et al., 2022) provide a new opportunity to track the detailed 3D change in subglacial water volume (Arthur

et al., 2025). To improve our understanding of subglacial lake systems in the Canadian Arctic, we develop a regression model to process the 2-m ArcticDEM strip product acquired between 2011 and 2021 (Porter et al., 2022). This regression model resolves the ice elevations constrained by multiple time-stamped DEMs. Compared to the traditional method that uses the elevation difference from two DEMs, this new method allows us to track the drainage and recharge events of active subglacial lakes in unprecedented detail.

In this study, the term 'subglacial lake' is defined as a water body that is primarily located underneath a glacier (Capps et al., 2010). Besides the classical subglacial lakes confined below a single glacier, we describe two other types of subglacial lakes for the first time, as well as their hydrological characteristics. Our discovery is sufficient to create the first subglacial lake inventory



focused on the Canadian Arctic. Observations of lake evolution can better constrain glacier dynamics and mass balance in this
rapidly melting region (Sharp et al., 2011; Noël et al., 2018; Hugonnet et al., 2021; Jakob and Gourmelen, 2023; Zemp et al.,
2025).

## 2  Materials and Methods

We use the Python-based Cryosphere And Remote Sensing Toolkit (CARST; Zheng et al., 2018, 2021) and the scikit-learn
package (Pedregosa et al., 2011) for the entire analysis workflow unless otherwise specified.

### 2.1  ArcticDEM elevation data

We focus on glacier and ice cap areas in the northern Canadian Arctic (ACN) and southern Canadian Arctic (ACS) subregions
defined by the Randolph Glacier Inventory (RGI) version 7.0 (RGI 7.0 Consortium, 2023). We collect the time-stamped Arc-
ticDEM strip Digital Elevation Models (DEMs) released in October 2022 (Porter et al., 2022), each with a spatial resolution
of 2 m. A total of 23,691 DEM strips intersect with the RGI glacier outlines, with observation dates ranging from 2011 to
2021. We resample each strip into a 15 m raster using bilinear interpolation and stack all strips to form a data cube. For each
spatial pixel in the data cube, we perform the modified Elevation Verification from Multiple DEMs (EVMD) algorithm (Zheng
et al., 2018) to flag potentially erroneous elevations. This modified algorithm adopts the DBSCAN method (Ester et al., 1996)
to identify clusters in the elevation-time space, with units in meters and years, respectively. Elevation measurements outside
any clusters are flagged as potentially erroneous elevations. We tested various DBSCAN parameters for the best performance.
The final algorithm sets *eps* (radius of neighborhood search) to 20 on a time-elevation space with units of year and meter,
respectively.  The  *minPts* parameter (minimum measurements to form a cluster) is set to 3 . We also flag measurements
with ArcticDEM bitmask labels indicating water, cloud, or edge as potentially erroneous elevations (Porter et al., 2022). We
excluded all flagged elevations from the rest of the analysis.

### 2.2  Broad scan for unusual glacier elevation change

The remaining measurements are fitted to a linear model pixel-by-pixel. We create a map of the long-term elevation change rate
($\frac{dh}{dt}$), which is the slope of the linear model, for the entire Canadian Arctic (Figure 2a–b). This map is available online (Zheng
et al., 2024) as one of the data sets used in the Glacier Mass Balance Intercomparison Exercise (GlaMBIE) project (Zemp
et al., 2025). We manually examine the map and mark areas with unusual patterns of elevation change, such as enhanced local
thinning and thickening within a glacier basin. We further review these areas by checking the stacked elevations together with
the optical images from the NASA/USGS Landsat 8, 9, and ESA Sentinel-2 satellites. Based on the conditions of land cover
from the optical images and the spatial characteristics of the elevation change signals, we classify these areas into one of the
following four categories:

1. **Active subglacial lake**: The objectives of this study; $\frac{dh}{dt}$ is distinct from the surrounding area and is coherent on the
   kilometer scale. Satellite images show ice surfaces without obvious changes.



2. **Glacier surge**: Coupled thinning and thickening signals adjacent to each other; thickening downstream and thinning upstream.

3. **Calving front**: Strong negative $\frac{dh}{dt}$ at the ice margin, adjacent to an ice-marginal or proglacial lake with surface elevation changing to lake level.

4. **New bedrock exposure**: $\frac{dh}{dt}$ is more neutral (closer to zero) than the surrounding area, and the unglacierized surface is recognized from optical images acquired from 2020 and onward.

We focus on areas classified as active subglacial lakes (37 in total) and reanalyze ArcticDEM elevation data as described in the next section.

## 2.3 Elevation change model for subglacial lakes

We reanalyze the same ArcticDEM data for each subglacial lake location. We coregister a total of 2,189 DEM strips over the lakes with the ICESat-2 ATL-6 measurements (Smith et al., 2023) over the neighboring off-ice land area using the Iterative Closest Point (ICP) algorithm provided by the NASA Ames Stereo Pipeline (Beyer et al., 2018). The vertical uncertainty for each coregistered DEM ($\sigma_i; i = 1, 2, ...n$ for the $N$-th DEM) is assigned as the root mean square error between the coregistered DEM and the reference ICESat-2 elevations (Zheng et al., 2018). The mean and median uncertainty of these coregistered DEMs are 1.24 and 0.68 m, respectively. The coregistered DEMs are again resampled into 15 m rasters, stacked, and verified using the EVMD algorithm described in the last section.

To model the ice elevation change due to lake activities, we manually review the temporal pattern of elevations for each lake and identify short-term elevation change events. Here, a short-term event is defined as the abrupt change ($> 10$ m) in ice surface elevation within one year relative to other times during 2011–2021. The regression model for the elevation change thus depends on how many short-term events we identify:

– If a lake does not show short-term events, we use a linear model:

$$h(t) = a_0 + a_1 t, \tag{1}$$

where $h$ is ice surface elevation and $t$ is time. The model parameters $a_0$ and $a_1$ are solved by the weighted least squares with weights set to $1/\sigma_i$.

– If a lake has one short-term event, we add a sigmoid term to the linear model:

$$h(t) = a_0 + a_1 t + \frac{L}{1 + e^{-k(t-t_0)}}, \tag{2}$$

where $L$, $k$, and $t_0$ are parameters that characterize the sigmoid. We normalize $t$ and $h$ prior to this nonlinear regression and solve the model parameters numerically with the weighted least squares. The reciprocal values of normalized uncertainty ($\sigma_h/\sigma_i$, where $\sigma_h$ is the standard deviation of $h$) are used as the weight of each data point.





– If a lake has two or more short-term events, or the elevation change does not fit the above models well, we use the

Gaussian Process (GP) regression (Rasmussen and Williams, 2005; Hugonnet et al., 2021; Pedregosa et al., 2011) to fit

the elevations. Both $t$ and $h$ are normalized before regression. We use data from Lake 14 (Mittie glacier, Figure 5e) to

validate and fine-tune kernel and hyperparameter combinations by comparing the model predictions with the elevation

change during Lake 14's drainage event in 2021. Finally, the selected GP model has a covariance matrix composed of

the sum of the normalized data variance and a rational quadratic kernel:

$$k(t_i, t_j) = \delta_{ij} \left( \frac{\sigma_i}{\sigma_h} \right)^2 + c \left[ 1 + \frac{(t_i - t_j)^2}{2\alpha l^2} \right]^{-\alpha}, \tag{3}$$

where $\delta_{ij}$ is a Kronecker delta that is 1 if $i = j$ and 0 otherwise. The hyperparameters $l$ (length scale) and $\alpha$ (scale

mixture) are fixed at 0.4 and 0.01, respectively, for all lakes. The only tunable hyperparameter for each lake during the

regression is the amplitude of the rational quadratic kernel $c$.

The selected regression model is applied to fit the elevation data pixel by pixel. To visualize lake extent, we derive a map of

elevation change during a short-term event (dH) for each lake (Figures 2c–d & 3). If the elevation change is fitted to a sigmoid

model, dH is analytically expressed by the sum of the sigmoid height ($L$) and the height change from the linear component

during the active event. We assume that the active event begins when the sigmoid term $L/(1 + e^{-k(t-t_0)})$ equals $0.01L$ for

recharge and $0.99L$ for drainage, and ends when the sigmoid term equals $0.99L$ for recharge and $0.01L$ for drainage (Figure

2d). Hence, dH can be written as:

$$dH = \sigma_h \left( L + \frac{2a_1 \ln 99}{k} \right). \tag{4}$$

If the ice elevation is modeled by the Gaussian Process, we simply assign dH as the maximum height gain or loss within one

year. We label the year for which the dH value is derived in Table 1 with an asterisk. The dH value is also reported in Table 1.

This step is omitted if a lake does not have a short-term event during the survey period.

## 2.4 Lake type classification

We classify the type for each subglacial lake using the modeled elevation data (dH map for lakes with short-term events; $\frac{dh}{dt}$

map for lakes without short-term events) and the optical satellite images described earlier based on the criteria below:

– If the elevation change signal extends to the ice margin and connects to open water visible in optical images, the lake is

assigned as type 3 (partial subglacial lake).

– If it is not type 3, and the lake is located in a valley topography where two or more glacier termini converge and coalesce,

the lake is assigned as type 2 (terminal subglacial lake).

– Otherwise, the lake is assigned as type 1 (classic subglacial lake).



## 2.5 Lake area and volume

We manually map lake areas using QGIS (https://www.qgis.org/) based on dH or $\frac{dH}{dt}$ maps by tracking places where ice elevation is clearly influenced by subglacial lakes. For type 3 lakes, we only map the lake area beneath the glacier ice. For a short-term drainage or recharge event, we assume that all volumetric change observed by the DEM analysis is due to the change in water volume (V), which can be calculated by

$$V = a_{\text{pixel}} \sum_{i=1}^{N} dH_i, \tag{5}$$

where $a_{\text{pixel}} = 225 \text{ m}^2$ is the area of a single pixel on the dH map, and N is the number of pixels within the lake area. The lake area and V are both reported in Table 1.

## 2.6 ITS_LIVE glacier velocities

To compare ice elevation change with glacier dynamics, we analyzed the scene-pair ice velocity maps provided by the NASA MEaSUREs ITS_LIVE project (Gardner et al., 2018, 2019). The data are accessed using NASA's itslive-py Python client and are visualized in panel f of Figures S1–S37.

## 2.7 ALOS-2 data

For a detailed case study, we use the Advanced Land Observing Satellite-2 (ALOS-2) PALSAR-2 sensor data for the interferometric synthetic aperture radar (InSAR) analysis. We select four PALSAR-2 images taken at the stripmap mode 3 (∼10-m spatial resolution) and form two interferograms for Milne Glacier, where Lakes 2, 3-a, 3-b, and 3-c are located. The first interferogram corresponds to the acquisition period between October 20 and November 17, 2015, and the second corresponds to the period between November 27 and December 25, 2018. The ArcticDEM mosaicked elevation data (Porter et al., 2023) is used to remove the phase change due to topography.

## 3 Results and discussion

### 3.1 Active subglacial lakes across the Canadian Arctic

We identified 37 active subglacial lakes across the Canadian Arctic (Figures 1–3, Table 1). Of these lakes, 32 have at least one short-term recharge or drainage period defined as an abrupt change (more than 10 m) in ice surface elevation within one year relative to the other time during the survey period of 2011–2021. For example, a short-term recharge event is identified for Lake 2 (at Milne Glacier, northern Ellesmere Island) when the ice surface elevation increased by more than 20 m in 2015 compared to relatively little elevation change during the pre- and post-event periods (Figure 4e). Some lakes had multiple short-term drainages during the survey period, which are specified in Table 1. The other five subglacial lakes (Lakes 5, 7, 25, 31, and 32)



do not have a short-term recharge or drainage, but are still considered active since the ice surface shows a significant decadal elevation change rate compared to the neighboring glacier region, which we attribute to constant lake recharge or drainage. We map the amount of elevation change (dH) during the selected short-term event (see Table 1); if a lake does not have one, we map the rate of elevation change (dH/dt) instead (Figure 3). These maps reveal the area with ice elevation change influenced by subglacial lakes, which we define as the lake area in this study (Table 1).

These lakes are widespread throughout the region, from 67°N (Penny Ice Cap, Baffin Island) to 82°N (Northern Ellesmere Island). However, only five (Lakes 15–17 and 33–34) are in the southern Canadian Arctic (Figure 1). Of the other 32 lakes in the northern Canadian Arctic, 20 are located in the North Ellesmere Ice Field and the Agassiz Ice Cap. We do not observe any elevation change related to subglacial hydrology at the proposed hypersaline lake site in the Devon Ice Cap, indicating that the lake is stable or non-existent (Rutishauser et al., 2018; Killingbeck et al., 2024). Based on optical images in the summer (Figures S1–S37, panels b) and a map of the average surface mass balance between 1958 and 2015 (Noël et al., 2018), we argue that all lakes reported in this study are either within the ablation zone or close to the boundary of the accumulation-ablation zone. This geographical distribution is consistent with the active subglacial lakes in the neighboring Greenland Ice Sheet (Fan et al., 2023; Gray et al., 2024a).

All lakes and short-term events are reported for the first time, except Lakes 14 and 19 in Manson Icefield (Gray et al., 2024b). Lake 14 is the largest among the reported lakes. Our analysis shows that it has an area of 48.5 km$^2$ and a drainage volume of 4.46 km$^3$ between late 2020 and early 2021 (Figures 3 and 5e). The glacier surface dropped vertically by up to 148 m during the drainage event. These estimates are consistent with the values reported in Gray et al. (2024b). In addition to the drastic drainage event in Lake 14, we identified four other lakes experiencing hydrological events that cause more than 100 m of elevation change during the study period. To compare, only two subglacial lakes in Iceland (Grímsvötn and E-Skaftár Cauldron) have reached this scale of elevation change during a jökulhlaup event (Björnsson, 2003; Livingstone et al., 2013). The largest active subglacial lake in Greenland (below the Flade Isblink ice cap) caused ∼70 m of elevation change from its drainage (Willis et al., 2015; Lenaerts et al., 2013), and for Antarctica, active subglacial lakes are associated with a much smaller elevation change, usually below 20 m (Lenaerts et al., 2013). Except Lake 14, the other subglacial lakes reported in this study have an area ranging from 0.3 to 15 km$^2$. Despite a smaller lake size, a short-term hydrological event can still cause a large change in surface elevations. All identified short-term events are associated with an elevation change of more than 10 m within one year, much higher than the average elevation change rate ($< 1$ m yr$^{-1}$) of the Canadian Arctic (Hugonnet et al., 2021; Zemp et al., 2025). In addition to the drainage event in Lake 14, the other short-term events are associated with a change in water volume between 0.003 and 0.6 km$^3$, or 0.003–0.6 Gt in mass assuming a water density of 1000 kg m$^3$.

## 3.2 Subglacial lake dynamics

Most of the reported lakes (30 of 37) undergo a slow recharge and fast drainage pattern (e.g., Figure 5c–f, h). These lakes recharge over the course of multiple years and drain within a year. Since ArcticDEM data do not contain elevations for every season, we cannot determine the exact subannual timing of all lake drainage events. However, our results indicate that the drainage can only take a few months, such as Lakes 2, 3a–c, and 4a–b, located in northern Ellesmere Island. Three Arctic-





DEM observations of lakes 4a and 4b captured the short-term draining phase in July 2020 (Figure 5h). Based on the sigmoid regression model, the entire event probably lasted only two to three months. For lakes 2 and 3a–c, although optical images (Landsat 8; Figure 4a–b) barely show surface change during the 2015 drainage (Lakes 3a–c) and recharge (Lake 2) event, the

210 InSAR image from ALOS-2 acquired between October 20 and November 17, 2015, shows decorrelated surface at Lakes 3a–c (Figure 4c) possibly due to a vertical elevation change of more than tens of meters, much larger than the submeter radar wavelength of ALOS-2. Decorrelation makes phase unwrapping impossible for a quantitative surface deformation analysis. Still, their presence suggests that these lakes are rapidly draining in October and November 2015, compared to the other InSAR pair acquired during fall 2018 (Figure 4d) when no short-term events are observed based on elevation data. These two examples are

215 consistent with the observations of Lake 14 in Gray et al. (2024b), for which the ICESat-2 data suggested a drainage period of 30–60 days. Similarly, subglacial lakes in the peripheral regions of the Greenland Ice Sheet typically drain within three to four months (Livingstone et al., 2022; Fan et al., 2023).

A short drainage duration may suggest the possible occurrence of a glacial lake outburst flood, but confirming it with observations of subglacial pathways is challenging. Nevertheless, our analysis reveals two interconnected lake systems, Lakes

3a–c and 4a–b, since their drainage is synchronized (Figures 4f and 5h). No human settlements are close to any of the lakes reported in this study, and therefore the risk of infrastructure damage and loss of life due to the drainage of these lakes is very low.

The duration of the drain-recharge cycle varies within the reported lakes from almost every year (Figure 5g) to once per decade (Figure 5c). In fact, it is possible that five subglacial lakes do not have short-term events simply due to a drain-recharge

cycle that is longer than our survey period. For example, the ice surface elevation at Lake 7 dropped approximately 10 meters more than the prediction of the linear regression model in 2021 (Figure 5b), implying a potential onset of short-term drainage after nine years of steady change. There are also cases with unclear or no cyclic behavior, such as short-term drainage without subsequent recharge (Lakes 4 and 6; Figure 5d and h) and short-term recharge without subsequent drainage (Lake 2; Figure 4e). These lakes may be associated with the formation of new inflow or outflow channels, which can transform a previously

stable subglacial lake into an active one (Livingstone et al., 2022). Detailed analysis, with extended temporal data coverage, will still be necessary to confirm whether these lakes were once stable but have turned active.

Contrary to common observations from the past literature that lake drainage affects glacier velocity (Stearns et al., 2008; Siegfried et al., 2016; Andersen et al., 2023), the analysis of the ITS_LIVE ice surface velocities (Gardner et al., 2018, 2019) shows that these subglacial lakes and short-term events do not appear to influence ice movement (Figures S1–S37, panel f).

The only possible case is Lake 17 when the ice speed reaches ∼30 m yr$^{-1}$ during drainage in the summer of 2015, compared to an ice speed of 0–20 m yr$^{-1}$ in other years (Figures S20f). This might suggest that most short-term drainage/recharge events are efficient through a channelized network, preventing the reduction of basal drag.

## 3.3 Subglacial lake classification

We classify these subglacial lakes into three types (Figure 6). The first type represents a "classic" subglacial lake confined

between a single glacier and the bedrock (Livingstone et al., 2022). The second type is found in locations where two glacier



termini converge and coalesce within a valley topography. We introduce a new term "terminal subglacial lake" for this type, indicating its location at the coalesced glacier terminus. The type-3 subglacial lake, or a "partial subglacial lake," includes an open water area at the ice margin as seen by joint analysis of optical images and elevation change. Among the 37 subglacial water bodies, eleven are classic subglacial lakes (Type 1), another eleven are terminus subglacial lakes (Type 2), and the remaining fifteen are partial subglacial lakes (Type 3).

Although this is the first time Type 2 lake (terminal subglacial) has been introduced, there have been a few lakes in the literature that can be classified as this type (Gray et al., 2024b, a). For a terminal subglacial lake to form, the ice at the coalesced terminus must be thinner than upstream to create an environment with low hydraulic potential. The valley side walls can thus provide good confinement for the water to accumulate. It is reasonable to assume similar water sources to classic subglacial lakes; that is, water can come from basal melt or surface melt that penetrates through the ice near or at the coalesced terminus. The compressional stress due to glacier convergence can create a heavily crevassed field as seen from the satellite images of these lakes (e.g., Lakes 13 and 19; Figures S16b and S22b), providing the necessary englacial routes to the lake (Willis et al., 2015). It should be noted that all the terminal subglacial lakes discovered so far have short-term events.

Type 3 subglacial lakes (partial subglacial) show a great variety in terms of open-water area. Some of them are mostly beneath the ice (e.g., Lakes 22 and 24), while others have a proglacial area that is several times bigger than the subglacial portion (e.g., Lakes 20 and 32). For the latter case, the overlying glacier is essentially a floating ice tongue with surface elevations largely influenced by lake level. Since these lakes extend to non-glacierized area, they can be recharged by sources such as direct precipitation and surface runoff, in addition to glacier melt. For partial subglacial lakes with drain-recharge cycles, the water likely drains through the damming ice when the hydraulic pressure is sufficient to expand the outflow channels. Outside of the Canadian Arctic, three subglacial lakes at Brady Glacier, Alaska, reported by Capps et al. (2010), can also be reclassified as Type 3 lakes as well since a small portion of these lakes are exposed along the glacier margin.

## 3.4    Correlation with regional mass balance

To understand whether subglacial lake activities have been influenced by recent ice loss in the Canadian Arctic, we perform a correlation analysis by comparing the number of short-term events each year with the annual glacier mass change published by the Glacier Mass Balance Intercomparison Exercise (GlaMBIE) project (Zemp et al., 2025). Since the event record for 2011 may be incomplete due to our search constraints, we focus on the data between 2012 and 2021 for comparison. The number of short-term subglacial events during these nine years has a significant correlation with the combined glacier mass change from the northern and southern Canadian Arctic ($r = -0.69$, $p$-value$= 0.039$) under a 5% significance level (Figure 7). This significance still holds if we exclude Type 3 lakes ($r = -0.74$, $p$-value$= 0.023$). The strong correlation may indicate that increased meltwater supply can quickly change the hydraulic conditions of these subglacial lakes, opening new outflow channels for efficient drainage. An elevated recharge rate can also result in a shorter drain-recharge cycle. The change in Lake 26 may imply this process: lake drainage (as indicated by ice elevation change) took place every year since 2019, when the regional glacier mass change switched from a slight gain (14 Gt) to a large loss (-105 Gt) for at least two years (Figures 5g and 7).



Considering the rapid changes and water volume associated with these lakes, it is therefore important to consider changes in subglacial hydrology when estimating the glacier mass balance, especially using geodesy-based methods, a process that has been inadequately incorporated due to a lack of understanding of the lake system. When applied to a regional or larger-scale analysis, these methods typically attribute the observed glacier surface height to the change in ice, firn, or snow volume, regardless of lake activities. However, our study shows that glaciers with an active subglacial lake can experience large variations in

mass change largely controlled by lake drain-recharge cycles. A better assessment should separate the mass balance of liquid water from that of ice. In doing so, glacier ice loss and subglacial lake activities can be more precisely observed and modeled. Since the change in glacier mass correlates with the frequency of lake activities, this separation process could become very important in the context of accelerated ice loss in the Canadian Arctic (Gardner et al., 2011; Lenaerts et al., 2013; Noël et al., 2018; Hugonnet et al., 2021; Zemp et al., 2025).

## 4 Conclusions and outlook

Our results present the first inventory of all known active subglacial lakes in the Canadian Arctic. It consists of 22 complete subglacial lakes (Types 1 and 2) and 15 partial subglacial lakes (Type 3). Even if we account for complete subglacial lakes, this number exceeds the number of active subglacial lakes in any other glacierized region, except Antarctica and the Greenland ice sheets (Livingstone et al., 2022; Fan et al., 2023). Furthermore, the Greenland ice sheet is at least 10 times larger than the

290 glacierized Canadian Arctic, but it has only 24 active lakes in the most updated inventory (Fan et al., 2023). This comparison indicates that many subglacial lakes are still likely to be discovered, which is consistent with the hydraulic predictions (Goeller et al., 2016).

Although we did not observe a clear link between lake activities and ice velocity changes, these lakes have the potential to influence the glacier mass balance. For partial subglacial lakes (Type 3), lake melting can cause additional loss of ice mass

that is typically overlooked by satellite geodesy, as observed in proglacial lakes (Zhang et al., 2023). As some of the lakes extend several kilometers from the ice margin to the glacier trunk, they can have a large contact area between ice and water, contributing to more hidden ice loss. In addition, the lake-terminating ice margin is prone to the calving process, allowing the retreat and mass loss of the calving front. Terminal subglacial lakes (Type 2) may also be vulnerable to these processes. Once the overlying ice melts or calves, they become partial subglacial lakes and are subject to instabilities related to ice-marginal

lakes. The detailed mechanism and significance of these lakes for glacier dynamics and mass balances are still unclear and require future fine-scale monitoring, followed by a better parameterization of the subglacial hydrology model.

*Code and data availability.* The ArcticDEM strip data set is available at https://doi.org/10.7910/DVN/C98DVS. The ICESat-2 ATL-6 data are available at https://doi.org/10.5067/ATLAS/ATL06.006. The RGI 7.0 glacier outlines for the Canadian Arctic are available at https://doi.org/10.5067/f6jmovy5navz. The ALOS-2 PALSAR-2 stripmap SAR images are available for query and purchase at https://alos-pasco.

com/en/offer/archive/. The derived data (excluding the InSAR results because we do not have authorization) necessary to reproduce the results and figures in this paper are available at https://doi.org/10.30238/TPIDR.DB_ISWCA/Dataset.



The CARST software is hosted on Github (https://github.com/ncu-cryosensing/CARST) and can be installed via PyPI (https://pypi.org/project/carst/). The scripts we use to generate the figures in this paper are available at https://github.com/whyjz/ac-subglacial-lakes with the Jupyter Book pages at https://whyjz.github.io/ac-subglacial-lakes/.

*Author contributions.* Conceptualization: WZ, WVW. Methodology: WZ. Investigation: WZ, TY. Software: WZ, TL. Validation, Visualization, & Writing - original draft: WZ, WVW, TL. Writing - review and editing: all authors.

*Competing interests.* Some authors are members of the editorial board of journal TC.

*Acknowledgements.* We thank the National Center for High-Performance Computing (NCHC) in Taiwan for providing the computational resources necessary for this study. Wesley Van Wychen acknowledges support from the Canada Foundation for Innovation (John Evan's Leadership Fund), the Ontario Research Fund, and Environment and Climate Change Canada (Climate Research Division). We acknowledge the ALOS4ice team for providing the ALOS-2/PALSAR-2 data (PI No. PER4A2N073).

*Funding sources*:

National Science and Technology Council of Taiwan, grants 112-2628-M-008-006 and 113-2116-M-008-024 (Whyjay Zheng)

Natural Sciences and Engineering Research Council of Canada (NSERC) Discovery Grant / Cette recherche a été financée par le Conseil de recherches en sciences naturelles et en génie du Canada (CRSNG), RGPIN-02443-2021 (Wesley Van Wychen)

European Union's Horizon 2020 research and innovation programme through the project Arctic PASSION, grant 101003472 (Tian Li)

Leverhulme Early Career Fellowship (ECF-2024-157) (Tian Li)



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



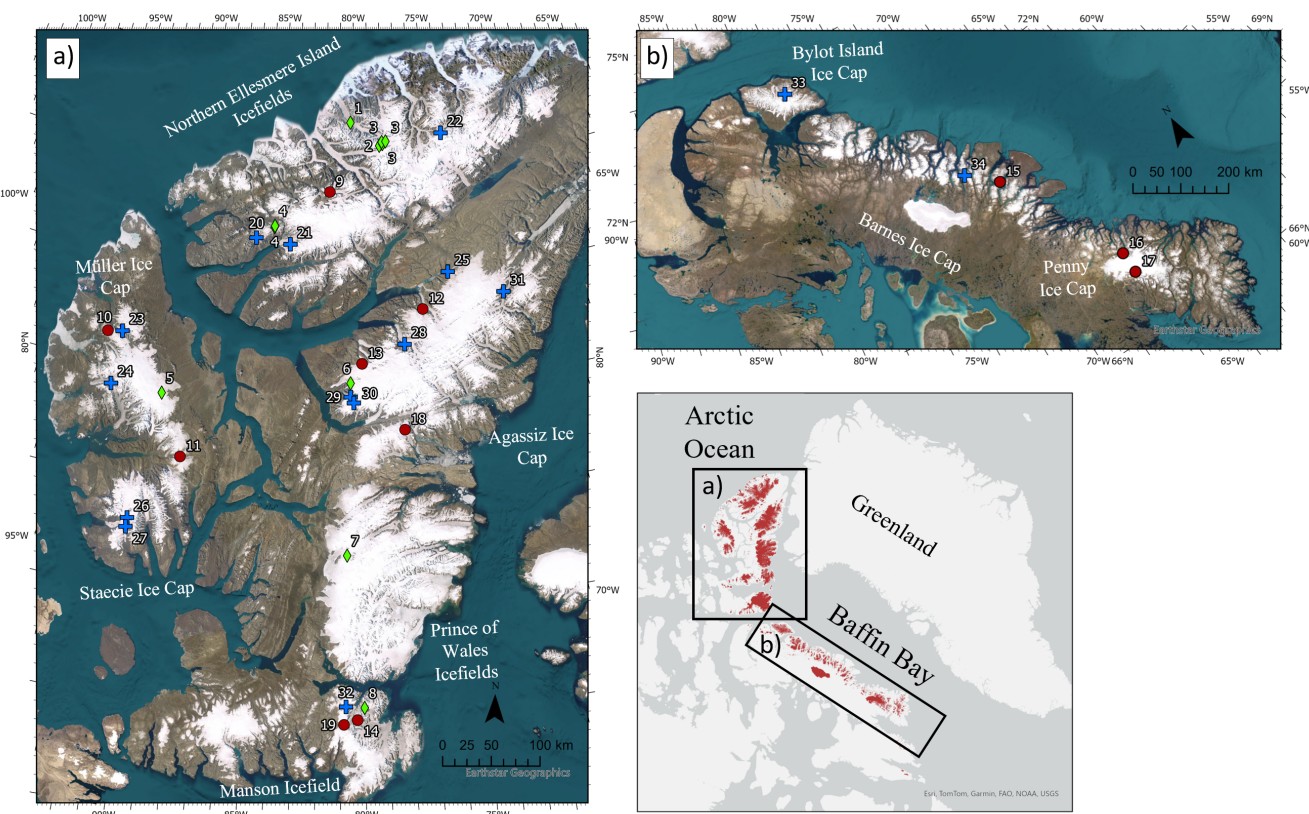

**Figure 1. The geographical locations of the active subglacial lakes in the Canadian Arctic.** (a) Lakes in Ellesmere and Axel Heiberg Islands (within the RGI region ACN); (b) Lakes in Baffin Island (within the RGI region ACS). The marker styles correspond to the subglacial water types defined in Figure 6: green diamonds are Type 1 lakes, red circles are Type 2 lakes, and blue crosses are Type 3 lakes. Lake numbers correspond to those in Table 1. Inset map shows the locations of (a) and (b) within the Canadian Arctic, with red areas indicating glacier mass.





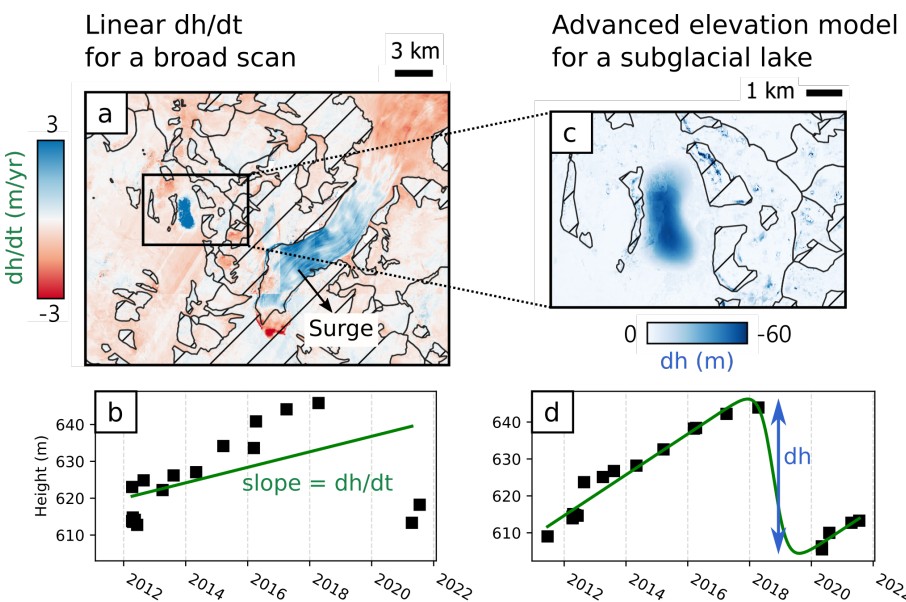

**Figure 2. Schematic of the analysis workflow.** (a) A map of linear elevation change rate is derived, highlighting signals such as glacier surge and subglacial lake. Hatched areas are unglacierized. (b) Example time series of the surface elevation at the subglacial lake site, fitted with a linear model. (c) The elevation data near the lake region are reprocessed using a selected regression model. A map of elevation change during the drainage event can be derived. (d) is similar to (b), but the elevations are fitted with a sigmoid model, with the indicated elevation change ($dh$) mapped in (c). See Sections 2.2–2.3 for more details.





**Figure 3. Elevation change of the active subglacial lakes.** The number of each panel corresponds to the lake number in Table 1. See Figure 1 for their geographical locations. Each panel is color-coded by the ice elevation change (dH) during the most significant drainage event between 2011 and 2022. Exceptions are lakes 2 and 13, which are color-coded by the dH of a recharge event. Lakes 5, 7, 25, 31, and 32 do not have any short-term events and are thus visualized by the linear elevation change rate (dH/dt) between 2011 and 2022. These lakes are classified into three types (Figure 6), grouped by a thick red line with labels on the left. Dark area in each map panel is unglacierized, defined by the RGI 7.0 glacier outlines (RGI 7.0 Consortium, 2023).



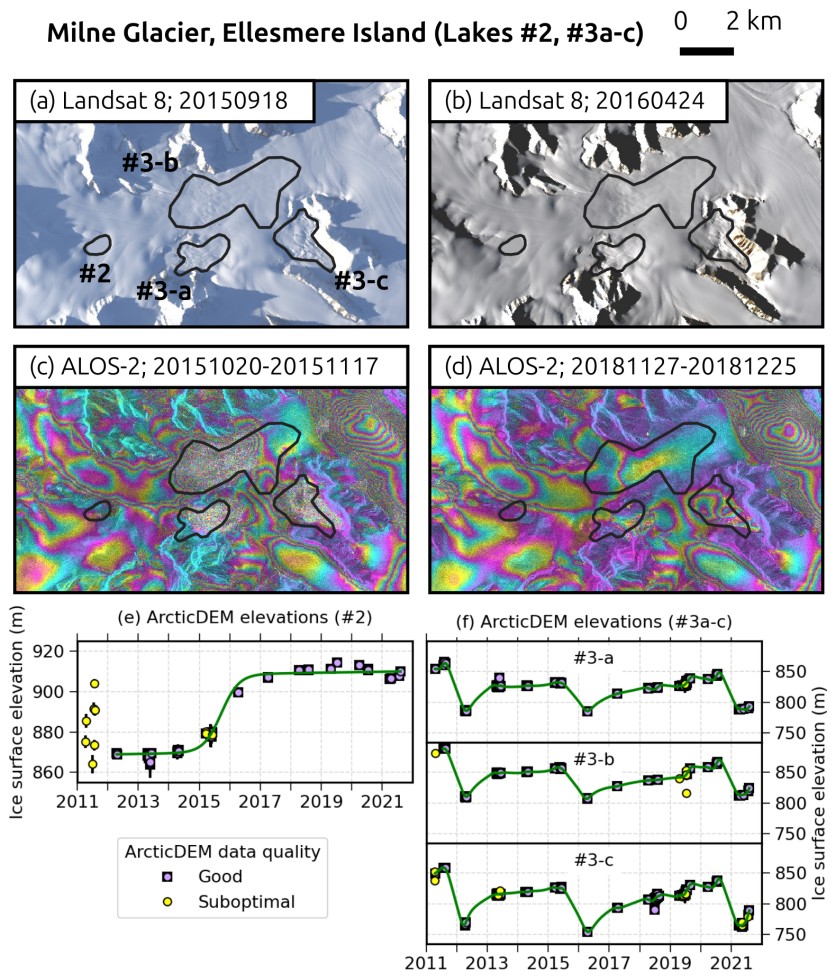

**Figure 4. Case study of four adjacent lakes.** (a) Landsat image acquired on September 18, 2015, with locations of Lakes 2, 3-a, 3-b, and 3-c. (b) Landsat image acquired on April 4, 2016, after a drainage event took place in late 2015 at lakes 3-a to 3-c. (c) ALOS-2 interferogram between October 20 and November 17, 2015. The significant phase change at lakes 3-a to 3-c indicates an ongoing drainage event. (d) ALOS-2 interferogram between November 27 and December 25, 2018, showing only a little surface change compared to 2015. (e) Ice surface elevation change over time for Lake 2. Purple squares are observations used for the regression model (green curve), and yellow circles are the flagged outliers excluded from the regression model. Vertical bars indicate the 2-$\sigma$ error of the observations. (f) Ice surface elevation change over time for lakes 3-a to 3-c.



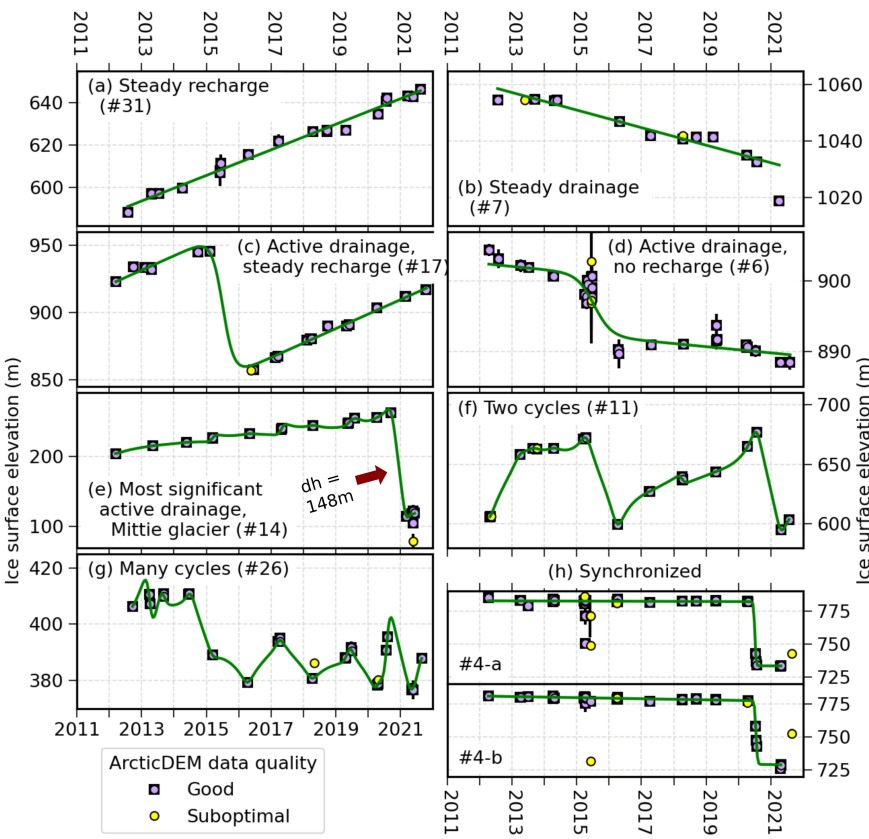

**Figure 5. Ice elevation change at the selected subglacial lake locations with various hydrological patterns.** Lake numbers correspond to Figures 1–3 and Table 1. The ICESat-2-aligned ArcticDEM elevations are plotted based on their data quality: good (purple squares) and suboptimal (yellow circles). The regression model (green lines) uses all the good data as input and has three kinds based on the hydrological patterns: linear (a–b), sigmoid (c–d; h), and Gaussian processes (e–g). Panel h shows Lakes 4-a and 4-b with synchronous drainage patterns. Vertical bars indicate the 2-$\sigma$ error of the observations. Plots for other lakes are available in Figures S1–S37.



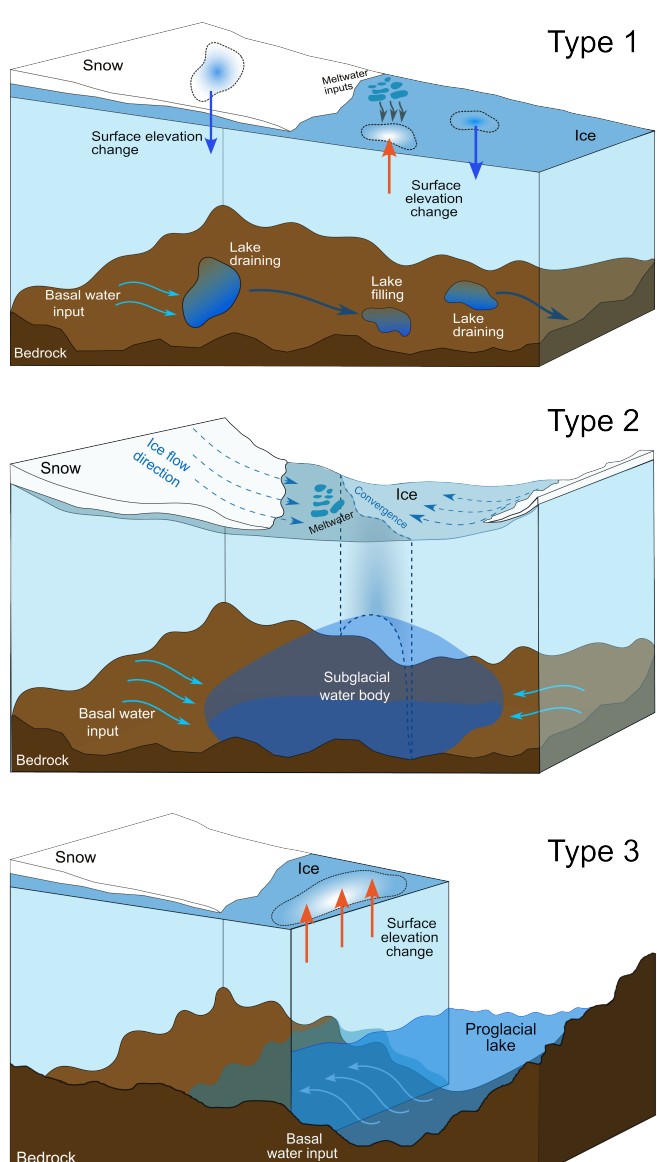

**Figure 6. Sketch showing different types of subglacial lakes described in this study.** A Type 1 lake (classic subglacial) is confined by a single glacier and the bedrock. Sketch based on Livingstone et al. (2022). A Type 2 lake (terminal subglacial) is found at the coalesced terminus where two glacier flows converge and accumulate in a constrained valley topography. A Type 3 lake (partial subglacial) includes an open water area that resembles a proglacial lake when viewed in the satellite imagery.



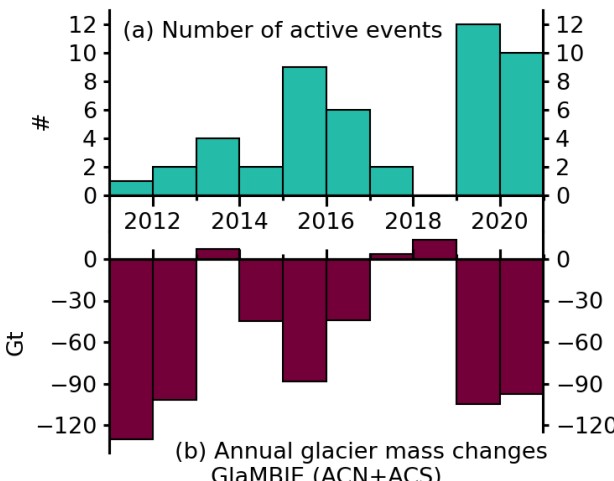

**Figure 7. Comparison between the subglacial event frequency and the regional glacier mass change.** (a) Total number of short-term events from the reported subglacial lakes each year. The record of short-term events in 2011–2012 may be incomplete due to searching constraints. (b) Annual glacier mass change from the GlaMBIE project (Zemp et al., 2025). The plot shows the combined mass balance of the two subregions in the Canadian Arctic (north and south) defined by RGI (RGI 7.0 Consortium, 2023). Each annual record represents one hydrological year from October 1 to September 30. For visualization purposes, we shift each record three months later and plot it starting on January 1 next year.



**Table 1. Active subglacial lakes identified in this study.** For each lake, this table describes its latitude (Lat), longitude (Lon), type (see Figure 6), years when a short-term drainage or recharge event took place, lake area (Area), vertical elevation change during the selected short-term event (dH), and volume change during the selected short-term event (V). The selected events are marked with an asterisk.

| # | Lat | Lon | Type | Event years | Area (km$^2$) | dH (m) | V (km$^3$) |
|---|---|---|---|---|---|---|---|
| 1 | 82.43 | -80.31 | 1 | 2016 | 1.0 | -28 | -0.014 |
| 2 | 82.21 | -78.38 | 1 | 2015 | 0.5 | +72 | 0.015 |
| 3-a | 82.23 | -78.16 | 1 | 2011*, 2015, 2020 | 1.8 | -104 | -0.080 |
| 3-b | 82.25 | -78.21 | 1 | 2011*, 2015, 2020 | 7.8 | -84 | -0.224 |
| 3-c | 82.25 | -77.94 | 1 | 2011*, 2015, 2020 | 2.6 | -101 | -0.099 |
| 4-a | 81.45 | -85.08 | 1 | 2020 | 2.0 | -54 | -0.053 |
| 4-b | 81.46 | -85.06 | 1 | 2020 | 0.4 | -50 | -0.008 |
| 5 | 79.82 | -90.34 | 1 | - | 2.9 | - | - |
| 6 | 80.04 | -80.47 | 1 | 2015 | 2.2 | -10 | -0.011 |
| 7 | 78.46 | -80.70 | 1 | - | 1.2 | - | - |
| 8 | 77.06 | -80.01 | 1 | 2016 | 0.9 | -17 | -0.007 |
| 9 | 81.80 | -81.69 | 2 | 2012, 2020* | 3.3 | -70 | -0.103 |
| 10 | 80.29 | -93.78 | 2 | 2017 | 1.4 | -82 | -0.065 |
| 11 | 79.27 | -88.94 | 2 | 2015, 2020* | 2.9 | -96 | -0.133 |
| 12 | 80.69 | -76.32 | 2 | 2014, 2019, 2020* | 0.8 | -82 | -0.018 |
| 13 | 80.22 | -79.84 | 2 | 2016 | 0.7 | +39 | 0.013 |
| 14 | 76.95 | -80.31 | 2 | 2020 | 48.5 | -148 | -4.461 |
| 15 | 69.94 | -69.48 | 2 | 2013, 2016, 2020* | 1.0 | -36 | -0.026 |
| 16 | 67.54 | -66.30 | 2 | 2012, 2015*, 2019, 2021 | 6.7 | -63 | -0.322 |
| 17 | 67.13 | -66.33 | 2 | 2015 | 8.9 | -98 | -0.521 |
| 18 | 79.60 | -77.72 | 2 | 2019 | 4.1 | -56 | -0.120 |
| 19 | 76.90 | -80.86 | 2 | 2019 | 1.3 | -70 | -0.060 |
| 20 | 81.34 | -86.14 | 3 | 2020 | 0.3 | -38 | -0.007 |
| 21 | 81.30 | -84.05 | 3 | 2019 | 0.8 | -15 | -0.010 |
| 22 | 82.28 | -74.14 | 3 | 2019 | 0.6 | -16 | -0.006 |
| 23 | 80.32 | -92.99 | 3 | 2019 | 3.6 | -54 | -0.146 |
| 24 | 79.82 | -93.03 | 3 | 2016 | 0.5 | -66 | -0.015 |
| 25 | 81.01 | -74.69 | 3 | - | 1.3 | - | - |
| 26 | 78.64 | -91.01 | 3 | 2014*, 2017, 2019, 2020 | 2.3 | -53 | -0.043 |
| 27 | 78.56 | -91.01 | 3 | 2016 | 0.4 | -11 | -0.003 |
| 28 | 80.38 | -77.48 | 3 | 2013*, 2015, 2019 | 1.6 | -63 | -0.055 |
| 29 | 79.91 | -80.47 | 3 | 2015, 2019* | 2.2 | -61 | -0.056 |
| 30 | 79.86 | -80.31 | 3 | 2019 | 5.4 | -108 | -0.358 |
| 31 | 80.76 | -71.64 | 3 | - | 15.1 | - | - |
| 32 | 77.07 | -80.77 | 3 | - | 3.3 | - | - |
| 33 | 73.33 | -78.18 | 3 | 2013 | 0.7 | -22 | -0.006 |
| 34 | 70.39 | -70.94 | 3 | 2013, 2015, 2019*, 2020 | 1.4 | -121 | -0.108 |