# Peer review of "Active subglacial lakes in the Canadian Arctic identified by multi-annual ice elevation changes"

_EGUsphere, 2025_

## Referee Comment (RC1)

This manuscript presents the first systematic inventory of active subglacial lakes in the Canadian Arctic based on ArcticDEM data from 2011–2021. The authors identify 37 lakes (35 of which are newly reported) and classify them into three categories, including two new types (terminal and partial subglacial lakes). The study provides quantitative estimates of lake area and volume changes, discusses recharge–drainage cycles, and reveals a significant negative correlation between lake activity and regional glacier mass balance. Overall, the paper delivers novel and valuable contributions to our understanding of subglacial hydrology in a rapidly changing region. Therefore, I recommend a minor revision of the paper.

**Major Comments**

1. The correlation analysis currently compares the annual number of subglacial lake events with the regional total glacier mass balance. While this yields a significant correlation, the choice of metrics may not be the most physically meaningful. Event counts do not capture the volume of water exchanged, and the regional mass balance may not reflect the local conditions of specific lake basins. A more convincing approach would be to compare the cumulative volume change of events ($\Delta V$) with the mass balance or runoff around the lakes?

2. While different regression models and error estimates are applied, potential under-detection of events and biases in water-volume estimates are not sufficiently discussed. I suggest the authors add a brief note in the Discussion or Conclusion to explicitly acknowledge these methodological constraints. In particular, it would be useful to clarify whether the number of DEM acquisitions varies between years, and if so, how this might influence the detection of short-term events and the apparent interannual variability in event counts. A short discussion of this potential bias would strengthen the robustness of the study.

Minor Comments

1. A clearer hierarchical structure would improve readability. In particular, separating the Methods, Data, and Results/Discussion sections more explicitly would help the

reader follow the workflow and findings more easily.

2.  The conclusion section restates results but could benefit from one or two sentences highlighting broader scientific implications or future directions, e.g., how satellite missions (ICESat-2, SWOT) might improve detection of similar events.

3.  In Figure 2, it would be helpful to indicate the locations corresponding to panels b and d. Currently, only one subglacial lake is shown in the figure; if these panels correspond to a different lake, please clarify this in the caption or text.2.

4.  In Figure 5, the ICESat-2-aligned ArcticDEM elevations are classified as "good" (purple squares) and "suboptimal" (yellow circles). Could the authors clarify how these categories are defined? Are they based on intrinsic DEM quality, on ICESat-2-to-DEM alignment performance, or some other criterion?

5.  Table 1 could be enhanced by indicating which lakes are newly reported to highlight the contribution, or the previous reported lakes can be labelled in Figure 1.

---

## Referee Comment (RC2)

Review of 'Active subglacial lakes in the Canadian Arctic identified by multi-annual ice elevation changes'. Whyjay et al.

This paper contributes new and valuable information on the existence, location, and multi-annual dynamic activity of subglacial lakes across Arctic Canada. To achieve this, the authors have optimized the use of available satellite imagery digital elevation models (DEMS) and lidar data to produce important metrics (area, timing of fill/drain events, magnitude of vertical displacements, and cyclical changes in subglacial lake water volumes) characteristic of the subglacial lakes identified. Results from this work have potential to guide future research towards understanding the response of subglacial lakes to future climate scenarios.

Scientific rigor of throughout this paper however is lacking. This paper exhibits poor writing style, inconsistent formatting, and very poor sentence structure; all of which obfuscate the intended meaning throughout. Many statements are left unsupported by proper referencing, inconsistencies amongst the figures exist in terms of style and background graphics exist, and figure captions are poorly written – see comments below.

This paper cannot be accepted in its current form.

INTRODUCTION

L4 & 5: provide uncertainties on the measurements of lake area and water volumes

L 6: need reference after "… yearly."

L7: " **to** the Canadian Arctic".

L7: The characterization 'nearly exclusive' is an oxymoron needs to be changed.

L7: Given the numerous subglacial lakes globally identified in Livingstone et al.,(2022), ie. 755, and the fact that a recent study by Gray et al. (2022) examined subglacial lakes that would meet the definitions of 'Type 1 and 2' as defined in this study, confirms that these type of subglacial lakes are not 'exclusive' to the Canadian Arctic.

L8: indicate coefficient of correlation here.

L8: is glacier loss 'accelerating' or just entered a step-wise shift to enhanced mass loss? If accelerating, this should be explained and referenced in the text.

L13: subglacial hydrological networks can be on the order of hundreds of kilometres long (Ehenfucht et al., 2024)

L 15-16: subglacial lake drainage events have been reported to increase glacier flow by hundredfold in Alaska (Kamb, et al., 1988), and higher for glaciers on Iceland.

L 19-20: Stearns, et al., (2008) is one of dozens of papers reporting on the influence of subglacial lake drainage events on glacier dynamics. This alone is not adequate evidence for suggesting subglacial lake drainage as a key mechanism for controlling glacier mass balance.

L 29: the word 'however' is not necessary here.

L33. The word 'partially' is not adequate. This region is heavily glacierized, particularly across the high Arctic (Arctic Canada North) where glaciers and ice caps cover ~50% of Ellesmere Island, the 10th largest island globally.

L 34: Please use the adopted terminology for identifying glaciers and ice caps in the Canadian Arctic, which is 'Arctic Canada North' and 'Arctic Canada South' (eg. RGI V7.0, Rounce et al., 2023).

L 34-36: when referring to the Canadian Arctic as whole, total mass change for this region is to be reported as a whole, ie. ACN + ACS, as it is displayed in Fig 7.

L 34-36: For which period (s) of time do these reported numbers of mass change apply? This value changes annually.

L 37: WRT the sentence 'The first discovery was a  hypersaline lake..'.  This sentence needs to be removed as there never was a hypersaline lake discovered on Devon, so it is incorrect to refer to it as such.

L 40: The fact that there are 2 subglacial lakes in close proximity of each other is not on its own a justification to suggest that more subglacial lakes across the Canadian arctic **must** exist.

L 45-46. Change '...*helped identify numerous...*"  to  '...has *been used to identify*...'

L 47: ...*limiting the potential discovery*... ' It would be worth noting here that the density of satellite coverage increases with latitude for polar orbiting satellites such as cryosat and icesat.

L46L: '... identify numerous subglacial lakes....'.  Where were these lakes 'discovered?

MATERIALS AND METHODS

L61: Add the remote sensing and RGI data to this intro.

L71: spell out DBSCAN acronym

L81: is visible crevassing not evident over active subglacial lakes? And is crevassing not an 'obvious change'?

L84: '… *conditions of land cover…',* do you mean glacier ice surfaces?

L 87 : do these features not exhibit crevassing around the grounded margin of the subglacial lake?

L 88: 'Satellite images..' should be written as 'Optical satellite image data'

L89: are these zones always close together? Or can they also be separated, if so by how far?

L94: why is the criteria for new bedrock exposure specifically state to be from '2020 onward'?

L 159: state basic specs for ALOS-2, PALSAR-2, ie. wavelength, perpendicular baseline.

- L183: Why is it necessary to 'argue' that the sub-glacial lakes in the Canadian high Arctic are located in the ablation zone? Shouldn't this be clear by comparing published elevations of the ELA with elevation of the subglacial lake?

- What is the relevance of the subglacial lakes residing in the ablation zone? Would it be possible for the subglacial lakes to exist above the ELA. Please expand on this.

- Are you referring to the elevation of the ice surface over the subglacial lake, or the elevation of the subglacial lake below the ice?

- It is recommended that the authors also explore the freely available NASA Operation IceBridge swath thickness data for these analyses. Where overlap exists the OIB data may provide important information on the ice geometry, basal conditions may help explain subglacial lake dynamics and surrounding hydrology.

L184:

- Using the post-2000 decadal ELA averages from insitu monitoring (Burgess and Danielson, 2022) to identify which glaciological zone the lakes reside in is more realistic than using the 1958-2015 ELA  by Noel et al., 2018.

   Reference: Burgess DO and Danielson BD (2022) Meighen ice cap: changes in geometry, mass, and climatic response since 1959. *Canadian Journal of Earth Sciences*, **59**, 884–896, ISSN 14803313 (doi: 10.1139/cjes-2021-0126)

L209: change '*barely show…' to 'show minimal surface change..'*

L219:  the subglacial pathway for Lake 14 was clearly revealed by Gray et al., 2024 and should be acknowledged as such.

CONCLUSIONS AND OUTLOOK

L 287: ' account for complete subglacial lakes...' change to 'account for complete subglacial lake **alone** '.

L: 289:

L: 290: Improper reference. Goeller et al., 2016 deals with Antarctic lakes not Greenland, upon which your comparison with the Canadian Arctic subglacial lakes is based.

L:291: in order to provide any substance to this claim, this statement must be backed up with evidence that the ice cap / ice sheet under discussion has morphological and glaciological characteristics that conform to conditions where subglacial lakes are known to exist.

L 294: What roles could subglacial lakes play towards ' ... *influence the glacier mass balance'*? Neither this study or Gray et al., 2024 noted any significant change in ice velocity during subglacial lake outflow events. Please be specific as to how the subglacial lake activity has affected mass balance, and support with references.

L294: '..., lake melting...', do you mean ablation of glacier ice due to contact with subglacial lake water? Please clarify

L295: the statement '...more hidden ice loss.' is highly speculative. We don't know anything about the level of contact between the water and ice, water temperature, water turbulence, ice accretion...

L297-298: Wording of the statement '...**allowing** the retreat and mass loss of the calving front.' needs to be changed. Calving of lake terminating ice fronts may however be **facilitated** by the presence of water at the margin.

L293-391: this paragraph reads like and introduction. Generalizations are being made without specific reference to what was observed. Please replace with a concise summary of what was observed, implications and how this work can be improved, including future monitoring strategies.

FIGURES

Figure 2.

- This lake (18) should be identified and matched to the mosaic in Fig 3.

- Hatch lines should be more narrowly spaced – they do not clearly define the non-glacierized areas.

- Why are non-glacierized areas depicted differently between figs 2 and 3?

- This sentence 'A map of the elevation change **can** be derived...' does not seem very effective or meaningful...

- ' using a selected (?) regression model ...' which model was used?

Figure 3.

- Usage of ' – ' is inconsistent between legends.

Figure 4: Many of the yellow (suboptimal) points appear to align with the valid points used in the analysis. Please explain how these suboptimal points differ from the 'good' points.

Fig. 4

-Polygons outlining Lakes 2 and 3a-c to are at such small scale that it is difficult see any differences between the 2015 and 2016 LandSat images

- Why do many of the yellow (suboptimal) points that align with the 'good' points?

-perhaps a different scale should be used for the point quality as the error bars are meaningless for most points.

- supplementary material for all figs?

- Please state

Fig. 7

- Spell out 7- axis titles in full.

- X axis titles are unclear due to location of 'year' labels. The labeling should definitely start at the first column. For fig 7 it should start at 2011, as it does in fig5

- Why do the x-axes for graphs in fig 7 show 2011-2021 but 2011-2022 for fig 5. Should these not match?

---

## Author Comment (AC1)

Dear Dr. Bonneau,

Thank you very much for the insightful comment and for sharing the work of Antropova et al.

For Lake 1, the grounding line information is indeed very relevant. We will add it to the revised manuscript. We digitized Figure 12 in the Antropova et al. paper and compared the grounding line positions with our lake outline. Our quick result (as shown in the attached screenshot; the grounding line colors correspond to those used in Figure 12b) indicates that the lake area, identified by its drainage event in 2016, is very close to the grounding line ($HL_{LW}$) in 2011. Based on ERS-2 data, the lake was located in the grounded zone of 2011, and some potential activities can be identified using SAR interferometry (as shown in the second attached screenshot, Figure 5b from Antropova et al., with the lake location labeled by a red square). We do not have the grounding line information from 2016; however, the lake was likely still within the grounded zone, as it would be challenging to depict a mechanism that lowered the ice surface elevation by ~20 m within one year if the ice had already achieved flotation.

Thank you for sharing the ocean mooring data. It is exciting to see such a temporal correlation between the Lake 3a-c drainage events and the thermal anomaly from nearby ocean water! A detailed analysis of individual lake(s) might be beyond the scope of this manuscript, but the subglacial lakes in Milne glacier are worth a follow-up project to understand the lake dynamics based on various data sets. We will be in touch to discuss future plans!

Whyjay Zheng (on behalf of the authors)

[Figure]

[Figure]

b) 2011 ERS-2: March 31, April 3 and May 6, 9

---

## Author Comment (AC2)

Dear Dr. Chang-Qing Ke, Dr. David Burgess, and editors:

Thank you for reviewing our manuscript and providing constructive feedback. We have replied to all comments below with proposed changes to the manuscript. The original review text is in gray, and our response is in green. Again, we appreciate your insightful ideas and your willingness to share them with us, which will improve the quality of our manuscript.

Whyjay Zheng (on behalf of the coauthors)
* * *
Reviewer 1 (Chang-Qing Ke)

> This manuscript presents the first systematic inventory of active subglacial lakes in the Canadian Arctic based on ArcticDEM data from 2011–2021. The authors identify 37 lakes (35 of which are newly reported) and classify them into three categories, including two new types (terminal and partial subglacial lakes). The study provides quantitative estimates of lake area and volume changes, discusses recharge–drainage cycles, and reveals a significant negative correlation between lake activity and regional glacier mass balance. Overall, the paper delivers novel and valuable contributions to our understanding of subglacial hydrology in a rapidly changing region. Therefore, I recommend a minor revision of the paper.

Thank you very much for your positive and insightful review!

> Major Comments
> 1. The correlation analysis currently compares the annual number of subglacial lake events with the regional total glacier mass balance. While this yields a significant correlation, the choice of metrics may not be the most physically meaningful. Event counts do not capture the volume of water exchanged, and the regional mass balance may not reflect the local conditions of specific lake basins. A more convincing approach would be to compare the cumulative volume change of events (ΔV) with the mass balance or runoff around the lakes?

As stated in the manuscript, the intended goal of this comparison is to understand "whether subglacial lake activities have been influenced by recent ice loss in the Canadian Arctic." All the existing mass balance estimates for this area have not excluded the mass loss due to subglacial lake activities; it is likely that the volume change due to subglacial events will be highly correlated with the mass balance around the lakes (or the mass balance of the hosting glaciers), making this correlation unconvincing. In addition, Lake #14 has a ΔV much larger than that of any other lakes in the inventory. Therefore, a comparison using ΔV would be highly skewed by this single lake. We believe that the correlation between two less dependent variables, the number of lake events and the regional mass balance, is a more valid argument to link mass balance and lake activities. (The total volume change during the short-term events every year is typically ~ 1% of the mass balance.)

> 2. While different regression models and error estimates are applied, potential underdetection of events and biases in water-volume estimates are not sufficiently discussed. I suggest the authors add a brief note in the Discussion or Conclusion to explicitly acknowledge these methodological constraints. In particular, it would be useful to clarify whether the number of DEM acquisitions varies between years, and if so, how this might influence the detection of short-term events and the apparent interannual variability in event counts. A short discussion of this potential bias would strengthen the robustness of the study.

We will add a paragraph in the discussion section covering the following three aspects:
1. Limits associated with our methodology for identifying the subglacial lakes and assessing the water volumes.
2. Interannual variability in DEM acquisitions and how it impacts the search for the lake events.
3. Potential bias associated with the water volume estimates of the lake events.

> Minor Comments
> 1. A clearer hierarchical structure would improve readability. In particular, separating the Methods, Data, and Results/Discussion sections more explicitly would help the reader follow the workflow and findings more easily.

We will separate the Materials and Methods section into the Data section and the Methods section for clarity. For the Results and Discussion section, we prefer to keep the current structure as it can help us better cross-link the results and our interpretations.

> 2. The conclusion section restates results but could benefit from one or two sentences highlighting broader scientific implications or future directions, e.g., how satellite missions (ICESat-2, SWOT) might improve detection of similar events.

We will add a brief discussion in Conclusions and Outlook on how current and future satellite missions will improve our ability to monitor subglacial lake activities. As the manuscript states at L45-48, ICESat-2 can greatly reduce the uncertainty of elevation change estimates but suffers from the gap between ground tracks. SWOT has the potential to detect subglacial lakes, but there will be two concerns:
1. SWOT's northern limit in coverage is 78N, meaning most of the Canadian Arctic North won't be covered.
2. SWOT does not include land ice as its primary mission goal.
Nevertheless, SWOT could be used to trace relevant signals, such as increased water outflow or volume at nearby proglacial lakes.

> 3. In Figure 2, it would be helpful to indicate the locations corresponding to panels b and d. Currently, only one subglacial lake is shown in the figure; if these panels correspond to a different lake, please clarify this in the caption or text.2.

Panels B and D are from the same lake shown in Panels A/C, but are sampled from different pixels. Thank you for pointing this out; the data used in Panels B and D should have been identical for better clarity. We will synchronize the data used in both panels and place a

marker in Panel C indicating the sampling point. We will also label the lake ID in this figure and add text labels for the different fitting methods in Panels B and D.

> 4. In Figure 5, the ICESat-2-aligned ArcticDEM elevations are classified as "good" (purple squares) and "suboptimal" (yellow circles). Could the authors clarify how these categories are defined? Are they based on intrinsic DEM quality, on ICESat2-to-DEM alignment performance, or some other criterion?

They are based on (1) the intrinsic DEM quality, provided as a bitmask raster for each ArcticDEM strip; (2) whether the elevations are flagged by the Elevation Verification from Multiple DEMs (EVMD) algorithm or not. Relevant description can be found in L69-77. We will add more detailed information to improve the clarity of this part in the revision.

> 5. Table 1 could be enhanced by indicating which lakes are newly reported to highlight the contribution, or the previous reported lakes can be labelled in Figure 1.

This is a good idea. We will create an additional column in Table 1 to indicate which lakes are newly found in our study, and which have been reported previously, with corresponding references.
* * *
Reviewer 2 (David Burgess)

> Review of 'Active subglacial lakes in the Canadian Arctic identified by multi-annual ice elevation changes'. Whyjay et al.
>
> This paper contributes new and valuable information on the existence, location, and multiannual dynamic activity of subglacial lakes across Arctic Canada. To achieve this, the authors have optimized the use of available satellite imagery digital elevation models (DEMS) and lidar data to produce important metrics (area, timing of fill/drain events, magnitude of vertical displacements, and cyclical changes in subglacial lake water volumes) characteristic of the subglacial lakes identified. Results from this work have potential to guide future research towards understanding the response of subglacial lakes to future climate scenarios.
>
> Scientific rigor of throughout this paper however is lacking. This paper exhibits poor writing style, inconsistent formatting, and very poor sentence structure; all of which obfuscate the intended meaning throughout. Many statements are left unsupported by proper referencing, inconsistencies amongst the figures exist in terms of style and background graphics exist, and figure captions are poorly written – see comments below.
>
> This paper cannot be accepted in its current form.

Thank you for your thorough and honest review. We will improve the overall clarity and paper structure in the revised manuscript to fully address the raised concerns. Please find our response below.

> INTRODUCTION
> L4 & 5: provide uncertainties on the measurements of lake area and water volumes

The maximum and minimum values are reported for these quantities in the abstract, which we believe are clearer and easier for readers to quickly get a broad overview of these lakes. We will add information in the discussion section regarding the limits and uncertainties of the lake area and volumes instead.

> L 6: need reference after "… yearly."

We prefer to avoid adding a reference citation in the abstract by adhering to the suggested abstract guidelines listed on the The Cryosphere website: "*reference citations should not be included in this section, unless urgently require*d". Instead, we will rephrase the sentence to emphasize the importance of the Canadian Arctic without explicitly referencing quantitative data.

> L7: " **to** the Canadian Arctic".

We will fix this.

> L7: The characterization 'nearly exclusive' is an oxymoron needs to be changed.

We will modify the wording here, taking into account the comment below.

> L7: Given the numerous subglacial lakes globally identified in Livingstone et al.,(2022), ie. 755, and the fact that a recent study by Gray et al. (2022) examined subglacial lakes that would meet the definitions of 'Type 1 and 2' as defined in this study, confirms that these type of subglacial lakes are not 'exclusive' to the Canadian Arctic.

We agree, and we will rephrase this sentence to clarify that the **classification** of this lake type is new, and some known lakes can be reclassified into this new type.

> L8: indicate coefficient of correlation here.

We will add the correlation coefficient here.

> L8: is glacier loss 'accelerating' or just entered a step-wise shift to enhanced mass loss? If accelerating, this should be explained and referenced in the text.

The glacier loss pattern is close to a step-wise shift to enhanced mass loss during the study period (2012-2022), see Figure 7. We will further explain this part in the Results and Discussion section.

> L13: subglacial hydrological networks can be on the order of hundreds of kilometres long (Ehenfucht et al., 2024)

We will review the provided and other relevant references and modify the statement as necessary.

> L 15-16: subglacial lake drainage events have been reported to increase glacier flow by hundredfold in Alaska (Kamb, et al., 1988), and higher for glaciers on Iceland.

Glacier surges indeed involve changes in subglacial hydrological conditions, but none of the past studies have clear evidence that water is released from a persistent subglacial lake during an active phase of a surge. We would be conservative about this suggested argument.

> L 19-20: Stearns, et al., (2008) is one of dozens of papers reporting on the influence of subglacial lake drainage events on glacier dynamics. This alone is not adequate evidence for suggesting subglacial lake drainage as a key mechanism for controlling glacier mass balance.

We will add other adequate references to this statement.

> L 29: the word 'however' is not necessary here.

We will fix this.

> L33. The word 'partially' is not adequate. This region is heavily glacierized, particularly across the high Arctic (Arctic Canada North) where glaciers and ice caps cover ~50% of Ellesmere Island, the 10th largest island globally.

We can see there is a disagreement about to what extent is adequate to be called "partially." To avoid this ambiguity, we will remove the word "partially."

> L 34: Please use the adopted terminology for identifying glaciers and ice caps in the Canadian Arctic, which is 'Arctic Canada North' and 'Arctic Canada South' (eg. RGI V7.0, Rounce et al., 2023).

We will review and modify these terms throughout the manuscript accordingly.

> L 34-36: when referring to the Canadian Arctic as whole, total mass change for this region is to be reported as a whole, ie. ACN + ACS, as it is displayed in Fig 7.

We will modify this sentence as per suggested.

> L 34-36: For which period (s) of time do these reported numbers of mass change apply? This value changes annually.

The data represent the glacier mass change from 2000 to 2023 (Zemp et al., 2025). We will add this information to the manuscript.

> L 37: WRT the sentence 'The first discovery was a hypersaline lake..'. This sentence needs to be removed as there never was a hypersaline lake discovered on Devon, so it is incorrect to refer to it as such.

We recognize this issue and will modify the sentence to ensure (1) the word "discovery" is not used (since the earlier work inferred the potential existence of the lake instead of discovering it), and (2) emphasize that the claimed subglacial lake was a misidentification according to Killingbeck et al. (2024).

> L 40: The fact that there are 2 subglacial lakes in close proximity of each other is not on its own a justification to suggest that more subglacial lakes across the Canadian arctic **must** exist.

Yes, you are correct. We will remove this statement.

> L 45-46. Change '…*helped identify numerous*…" to '…*has been used to identify*…'

We will change it accordingly.

> L 47: …*limiting the potential discovery*… ' It would be worth noting here that the density of satellite coverage increases with latitude for polar orbiting satellites such as cryosat and icesat.

Thank you for the suggestion. We will consider including the information in the revised manuscript.

> L46L: '*… identify numerous subglacial lakes*….'. Where were these lakes 'discovered?

We will expand this sentence by providing the suggested details.

> MATERIALS AND METHODS
> L61: Add the remote sensing and RGI data to this intro.

We will redesign the intro section and add the suggested information as we split this section into the materials section and methods section (see minor comment #1 from Reviewer 1).

> L71: spell out DBSCAN acronym

We will add the full name of the DBSCAN method.

> L81: is visible crevassing not evident over active subglacial lakes? And is crevassing not an 'obvious change'?

Crevassing is not exclusively linked to a subglacial lake origin. Compared to altimetry data, crevassing as seen in optical images can also be inconsistent over time if heavy snow cover is present. To further clarify, we will mention that "obvious change" is large-scale and not associated only with crevasses.

> L84: '… *conditions of land cover…*' , do you mean glacier ice surfaces?

We mean the land cover at places where we observe the unusual elevation changes. There are three possibilities: glacier surface (glacier surge / subglacial lake), water (ice marginal lakes / calving front / Type 3 subglacial lakes), and unglacierized terrain (new bedrock exposure).

We will modify the relevant text in the manuscript to improve clarity.

> L 87 : do these features not exhibit crevassing around the grounded margin of the subglacial lake?

Crevassing is present at most (though not always) of the lake margins. The changes in these crevasse fields before and after an active event, however, are less obvious than the changes in glacierized area, which we use to distinguish the other three categories. We will modify the text to better reflect this idea.

> L 88: 'Satellite images..' should be written as 'Optical satellite image data'

This will be modified as per suggested.

> L89: are these zones always close together? Or can they also be separated, if so by how far?

We will change the wording to: These zones must be adjacent to each other for us to classify the signal as a glacier surge.

> L94: why is the criteria for new bedrock exposure specifically state to be from '2020 onward'?

Choosing this specific year is for convenience in reviewing the optical images. To clarify our strategy, we will change the sentence to "dh/dt is more neutral (closer to zero) than the surrounding area, and the new unglacierized surface can be recognized from optical images. We compare the images acquired after 2020 and during 2013-2015 to identify new unglacierized surfaces."

> L 159: state basic specs for ALOS-2, PALSAR-2, ie. wavelength, perpendicular baseline.

We will add specs and baseline information.

> - L183: Why is it necessary to 'argue' that the sub-glacial lakes in the Canadian high Arctic are located in the ablation zone? Shouldn't this be clear by comparing published elevations of the ELA with elevation of the subglacial lake?

We will include the comparison between the ELA and the lake elevation and update the relevant description.

> - What is the relevance of the subglacial lakes residing in the ablation zone? Would it be possible for the subglacial lakes to exist above the ELA. Please expand on this.

Our methods can only detect active subglacial lakes. In Greenland Ice Sheet, stable subglacial lakes tend to be located above the ELA (Bowling et al., 2019; Fan et al; 2023). Hence, it would be worthwhile to describe the spatial distributions of the subglacial lakes reported in this study in terms of glacier zones. We will explain this relevance more clearly in the revised draft. We will explain this relevance more clearly in the revised draft.

> - Are you referring to the elevation of the ice surface over the subglacial lake, or the elevation of the subglacial lake below the ice?

For L181, we refer to the ice surface elevation. We will clarify this in the revised draft.

> - It is recommended that the authors also explore the freely available NASA Operation IceBridge swath thickness data for these analyses. Where overlap exists the OIB data may provide important information on the ice geometry, basal conditions may help explain subglacial lake dynamics and surrounding hydrology.

This is a good idea, and we will explore the OIB data in our follow-up research for these subglacial lakes.

> L184:
> - Using the post-2000 decadal ELA averages from insitu monitoring (Burgess and Danielson, 2022) to identify which glaciological zone the lakes reside in is more realistic than using the 1958-2015 ELA by Noel et al., 2018.
> Reference: Burgess DO and Danielson BD (2022) Meighen ice cap: changes in geometry, mass, and climatic response since 1959. Canadian Journal of Earth Sciences, 59, 884–896, ISSN 14803313 (doi: 10.1139/cjes-2021-0126)

We will integrate these field measurements of ELA to guide our inferences of where the ablation/accumulation zones begin/end.

> L209: change 'barely show…' to 'show minimal surface change..'

We will change it accordingly.

> L219: the subglacial pathway for Lake 14 was clearly revealed by Gray et al., 2024 and should be acknowledged as such.

We will modify the sentence to acknowledge the successful detection by Gray et al.

> CONCLUSIONS AND OUTLOOK
> L 287: ' account for complete subglacial lakes…' change to 'account for complete subglacial lake alone '.

We will change it accordingly.

> L: 289:
> L: 290: Improper reference. Goeller et al., 2016 deals with Antarctic lakes not Greenland, upon which your comparison with the Canadian Arctic subglacial lakes is based.

Thank you for pointing this out. We will remove this argument regarding Goeller et al. (2016).

> L:291: in order to provide any substance to this claim, this statement must be backed up with evidence that the ice cap / ice sheet under discussion has morphological and glaciological characteristics that conform to conditions where subglacial lakes are known to exist.

This comment will be addressed as we plan to remove the associated argument (see response for L290).

> L 294: What roles could subglacial lakes play towards ' ... influence the glacier mass balance' ? Neither this study or Gray et al., 2024 noted any significant change in ice velocity during subglacial lake outflow events. Please be specific as to how the subglacial lake activity has affected mass balance, and support with references.

Our hypothesis is that these lakes can contribute to ice loss by melting even without a change in ice velocity. However, we agree that this is a new idea to test, and there are no prior studies that have examined it. Therefore, we will address this comment by being conservative in the writing. We will state "these lakes have the potential to influence the glacier mass balance as indicated by the correlation found in this study."

> L294: '…, lake melting…' , do you mean ablation of glacier ice due to contact with subglacial lake water? Please clarify

Yes, it is referring to the ablation of glacier ice due to contact with subglacial lake water. We will change the text to better reflect this idea.

> L295: the statement '…more hidden ice loss.' is highly speculative. We don't know anything about the level of contact between the water and ice, water temperature, water turbulence, ice accretion…

We will remove this statement.

> L297-298: Wording of the statement '…allowing the retreat and mass loss of the calving front.' needs to be changed. Calving of lake terminating ice fronts may however be facilitated by the presence of water at the margin.

We will change the wording as suggested.

> L293-391: this paragraph reads like and introduction. Generalizations are being made

> without specific reference to what was observed. Please replace with a concise summary of what was observed, implications and how this work can be improved, including future monitoring strategies.

Thank you for the general comment regarding this last paragraph. To improve, we plan to merge the original paragraph into Results and Discussion (with the other associated comments addressed), specifically the section about the glacier mass balance. We will write a new concluding paragraph summarizing what we have found regarding new lakes, the significance of this study, and future missions that can be used to further improve lake detection.

> FIGURES
> Figure 2.
> - This lake (18) should be identified and matched to the mosaic in Fig 3.
> - Hatch lines should be more narrowly spaced – they do not clearly define the nonglacierized areas.
> - Why are non-glacierized areas depicted differently between figs 2 and 3?
> - This sentence 'A map of the elevation change can be derived…' does not seem very effective or meaningful…
> - ' using a selected (?) regression model …' which model was used?

Thank you very much for the comments. We will incorporate all of the suggested improvements here for Figure 2.

> Figure 3.
> - Usage of ' – ' is inconsistent between legends.

Thank you for spotting this. It will be fixed in the revision.

> Figure 4: Many of the yellow (suboptimal) points appear to align with the valid points used in the analysis. Please explain how these suboptimal points differ from the 'good' points.

They are based on (1) the intrinsic DEM quality, provided as a bitmask raster for each ArcticDEM strip; (2) whether the elevations are flagged by the Elevation Verification from Multiple DEMs (EVMD) algorithm or not. A relevant description can be found in L69-77. We will make a better connection between the description and the figure caption.

> Fig. 4
> -Polygons outlining Lakes 2 and 3a-c to are at such small scale that it is difficult see any differences between the 2015 and 2016 LandSat images
> - Why do many of the yellow (suboptimal) points that align with the 'good' points?
> -perhaps a different scale should be used for the point quality as the error bars are meaningless for most points.
> - supplementary material for all figs?
> - Please state

These lakes did not change their surface appearance much during the event as seen from the optical images. We will redesign the figure (e.g., zooming in) and make sure readers can fully explore the details at the scale of each individual lake.

For the suboptimal points, they are based on (1) the intrinsic DEM quality, provided as a bitmask raster for each ArcticDEM strip; (2) whether the elevations are flagged by the Elevation Verification from Multiple DEMs (EVMD) algorithm or not. A relevant description can be found in L69-77. We will add more detailed information to improve the clarity of this part in the revision. We will also remove the error bars in these plots, as they can be confusing as suggested.

> Fig. 7
> - Spell out 7- axis titles in full.
> - X axis titles are unclear due to location of 'year' labels. The labeling should definitely start at the first column. For fig 7 it should start at 2011, as it does in fig5
> - Why do the x-axes for graphs in fig 7 show 2011-2021 but 2011-2022 for fig 5. Should these not match?

Due to the search strategy, the subglacial events are very likely incomplete between 2021 and 2022, and we decided not to include this year for comparison. We will make sure this is clearly stated in the updated manuscript.

We will update the figure axes (including x-axis labels and y-axis names) as suggested.

**References**

Bowling, J. S., Livingstone, S. J., Sole, A. J., & Chu, W. (2019). Distribution and dynamics of Greenland subglacial lakes. Nature Communications, 10(1), 2810. https://doi.org/10.1038/s41467-019-10821-w

Fan, Y., Ke, C.-Q., Shen, X., Xiao, Y., Livingstone, S. J., & Sole, A. J. (2023). Subglacial lake activity beneath the ablation zone of the Greenland Ice Sheet. The Cryosphere, 17(4), 1775–1786. https://doi.org/10.5194/tc-17-1775-2023

Noël, B., van de Berg, W. J., Lhermitte, S., Wouters, B., Schaffer, N., and van den Broeke, M. R. (2018). Six Decades of Glacial Mass Loss in the Canadian Arctic Archipelago, Journal of Geophysical Research: Earth Surface, 123, 1430–1449, https://doi.org/10.1029/2017JF004304

Zemp, M., Jakob, L., Dussaillant, I., Nussbaumer, S. U., Gourmelen, N., Dubber, S., A, G., Abdullahi, S., Andreassen, L. M., Berthier, E., Bhattacharya, A., Blazquez, A., Boehm Vock, L. F., Bolch, T., Box, J., Braun, M. H., Brun, F., Cicero, E., Colgan, W., … Zheng, W. (2025). Community estimate of global glacier mass changes from 2000 to 2023. Nature, 639(8054), 382–388. https://doi.org/10.1038/s41586-024-08545-z